# Antimicrobial Resistance of *Listeria monocytogenes* Strains Isolated in Food and Food-Processing Environments in Italy

**DOI:** 10.3390/antibiotics13060525

**Published:** 2024-06-03

**Authors:** Antonio Rippa, Stefano Bilei, Maria Francesca Peruzy, Maria Grazia Marrocco, Patrizia Leggeri, Teresa Bossù, Nicoletta Murru

**Affiliations:** 1Department of Veterinary Medicine and Animal Production, University of Naples “Federico II”, Via Delpino 1, 80137 Naples, Italy; antonio.rippa@unina.it (A.R.); murru@unina.it (N.M.); 2Department of Food Microbiology, Istituto Zooprofilattico Sperimentale of Regions Lazio and Toscana “Mariano Aleandri”, Via Appia Nuova 001411, 00178 Rome, Italy; stefanobilei@gmail.com (S.B.); mariagrazia.marrocco@izslt.it (M.G.M.); patrizia.leggeri@izslt.it (P.L.); teresa.bossu@izslt.it (T.B.); 3Task Force on Microbiome Studies, University of Naples Federico II, 80138 Naples, Italy

**Keywords:** antimicrobic resistance, *Listeria monocytogenes* serovars, foods, public health

## Abstract

*Listeria monocytogenes*, along with various other pathogenic bacteria, may show resistance against a broad spectrum of antibiotics. Evaluating the extent of resistance in harmful microorganisms like *Listeria monocytogenes* holds significant importance in crafting novel therapeutic strategies to mitigate or combat the rise of infections stemming from antibiotic-resistant bacteria. The present work aims to investigate the occurrence of antimicrobial resistance among *Listeria monocytogenes* strains in meat products (n = 173), seafood (n = 54), dairy products (n = 19), sauces (n = 2), confectionary products (n = 1), ready-to-eat rice dishes (n = 1), and food-processing environments (n = 19). A total of 269 *Listeria monocytogenes* strains belonging to eight different serovars were tested against 10 antimicrobials. In the classes of antibiotics, most of the strains were resistant antibiotics belonging to the family of β-lactams (92.94%). High proportions of *L. monocytogenes* isolates were resistant to oxacillin (88.48%), followed by fosfomycin (85.87%) and flumenique (78.44%). The lowest level of resistance was observed against gentamycin (1.49%). A total of 235 strains (n = 87.36%) showed a profile of multidrug resistance. In conclusion, a high occurrence of resistant and multidrug-resistant strains of *Listeria monocytogenes* was observed among the examined serotypes isolated from different food sources. This understanding enables the adoption of suitable measures to avert contamination and the spread of resistant bacteria via food.

## 1. Introduction

*Listeria* (L.) species (spp.) are short Gram-positive rods that are non-sporulating, facultative anaerobes, catalase positive, and oxidase negative, and can grow under a wide range of conditions. The optimum growth temperature for *Listeria* spp. is 37 °C, but these bacteria can grow in a range between 4 °C and 45 °C and can survive in matrices with a pH between 4.7 and 9.6 [1]. *Listeria* spp. are ubiquitous bacteria commonly found in soil, water, sewage, decaying vegetation, and in animals such as birds and ruminants [2].

The number of identified species belonging to the genus *Listeria* has recently increased, and 17 in total have been identified. Among *Listeria* species, *L. monocytogenes* is considered the major causative agent responsible for diseases in humans [3].

Strains of *L. monocytogenes* can be classified into 13 serotypes (1/2a, 1/2b, 1/2c, 3a, 3b, 3c, 4a, 4ab, 4b, 4c, 4d, 4e, 7) according to their somatic and flagellar antigens [4]. The serotypes can also be regrouped into four lineages (I–IV) that differ genetically. Lineages associated with human infection are I and II [5]. The serotypes most isolated are 1/2b and 4b (lineage I) and 1/2a (lineage II). However, serotype 4b can also be included in lineage III or IV [6].

The primary route of transmission for listeriosis is through contaminated food, which is estimated to account for approximately 99% of cases [7]. The disease in humans ranges from mild to severe and can have a fatal outcome. People at risk from more serious illnesses are children, pregnant women, elderly people, and immunocompromised people, in which even a low level of *Listeria* in foods may cause serious illnesses such as septicemia, consequent meningitis, and, in cases of pregnant women, fetal infections and possible abortion [3]. Epidemiological source tracing is also very difficult because the signs of listeriosis appear after a long incubation time [8]. This incubation time differs depending on the clinical form: the longest period is registered in pregnancy cases (median 27.5 days), followed by the central nervous system infection (median 9 days), sepsis (median 2 days), and febrile gastrointestinal disease (median 24 h) [9].

The Food and Drug Administration (FDA) points out a stable incidence of listeriosis between the years 2004 and 2013 in the USA, while the European Food Safety Authority (EFSA) makes evidence that in the European Union, the incidence of listeriosis increased during the period between the years 2010 and 2014, probably due to the increase in the number of susceptible people, such as the elderly or immunocompromised people [10]. *L. monocytogenes* is a ubiquitous bacterium, a characteristic that benefits it in its diffusion, particularly in those foods that do not need cooking before consumption, known as ready-to-eat (RTE), which in recent years have had an increase in diffusion due to lifestyle changes. In 2022, 2738 cases of invasive listeriosis in humans were reported, of which 1330 were hospitalized and 286 died in Europe [11]. In 2022, the highest occurrence of Listeria was found in ‘fish and fishery products’ and ‘products of meat origin’ [11].

The treatment of listeriosis is difficult because *Listeria* spp. is an intracellular pathogen; some antibiotics are active in vitro but not in vivo, where they have only a bacteriostatic effect [12]. An ideal active antibiotic against *Listeria* spp. should have receptors that can bind the penicillin-binding protein (PBP-3). Therefore, antimicrobial agents belonging to the beta-lactams class are recommended [12].

*L. monocytogenes*, as well as other pathogenic bacteria, can exhibit resistance to a wide range of antibiotics [13]. Antibiotic-resistance (ABR) is a global health concern declared a major global health threat of the 21st century by leading regulatory authorities such as the International Monetary Fund (IMF), the World Bank, and the World Health Organization (WHO) [14]. It is estimated that 70,000 people die annually from infections caused by antibiotic-resistant microorganisms (ARM), and this trend is set to increase over the years.

*Listeria monocytogenes* represents a constant challenge for the food industry, health regulatory officials, and consumers since it remains one of the most virulent foodborne pathogens for vulnerable people [15]. Antibiotic resistance plays an important role in the increased incidence of different bacterial infections [16], and therefore, a high occurrence of *L. monocytogenes*-resistant strains in food could pose a serious risk to public health. The presence of resistance strains in foods has been recently investigated in Romania, China, and Egypt [7,15,17]; however, recent data on this phoneme’s extent in Europe are limited. The assessment of the degree of resistance across different geographical regions in pathogenic microorganisms such as *Listeria monocytogenes* is of primary importance for safeguarding public health and for the development of new plans to prevent or counter the emergence of infections caused by antibiotic-resistant bacteria [16].

Therefore, the aim of this work was to evaluate the occurrence of *Listeria monocytogenes*-resistant strains isolated in food in order to evaluate the status of the resistance against antimicrobics and possible countermeasures.

## 2. Results

The 269 strains analyzed of *Listeria monocytogenes* showed 88 different patterns; only 1 strain was sensitive against all the antibiotics (0.37%), whilst 235 strains were resistant against three or more classes of antibiotics (n = 87.36%) and therefore were classified as MDR.

In the classes of antibiotics, most of the strains were resistant to β-lactams (n = 250, 92.94%), fosfomycin (n = 233, 86.62%), and quinolones (n = 211, 78.44%) (Figure 1).

The highest level of resistance was observed against oxacillin (n = 238, 88.48%), followed by fosfomycin (n = 231, 85.87%), and flumenique (n = 211, 78.44%). Whilst the lowest level of resistance was observed against gentamycin (n = 4, 1.49%) (Table 1).

In serovars excluding those with a limited number of collected strains (1/a, 3a, 3b, 4d/4e, 4e), resistance to oxacillin, fosfomycin, and flumenique was extremely high in serovar 1/2a (oxacillin = 79, 84.95% fosfomycin = 73, 78.49% and flumenique = 75, 80.65%), 1/2b (oxacillin = 45, 93.75% fosfomycin = 46, 95.83% and flumenique = 36, 75.00%), 1/2c (oxacillin = 76, 86.36%, fosfomycin = 78, 88.64% and flumenique = 70, 79.55%), and 4b/4e (oxacillin = 26, 100.00% fosfomycin = 21, 80.77% and flumenique = 21, 80.77%). A total of 26 (54.17%) strains belonging to serovar 1/2b and 15 (57.69%) belonging to 4b/4e were resistant to ampicillin (Table 1). Moreover, almost half of the strains belonging to serovars 1/2b (n = 22, 45.83%) and 1/2c (n = 43, 48.86%) were resistant against lincomycin.

Strains belonging to serovars 1/2a and 4b/4e were highly resistant to trimetoprim–sulfametoxassole (1/2a= 42, 45.16% and 4b/4e = 10, 38.46%). Seventy-eight strains belonging to 1/2a (83.87%), eighty strains belonging to serovar 1/2c (90.9%), and forty-three strains belonging to serovar 1/2b (89.58%) were MDR (Figure 2). No correlation has been found between resistance and serotypes (*p* > 0.05).

In food categories, most of the strains resistant to ampicillin were isolated from dairy products (n = 12, 63.16%) and the environment (n = 12, 63.16%). A total of 94.74% (n = 18) of the strains isolated from dairy products and 85.55% (n = 148) of the strains isolated from meat products were resistant to oxacillin. In meat products, 79.19% (n = 137) of the strains were also resistant to flumequine, 43.93% (n = 76) to ampicillin, and 39.31% (n = 68 strains) to sulfamethoxazole–trimethoprim. In seafood, 90.74% (n = 49) of the strains were resistant against oxacillin, 75.92% (n = 41) to flumequine, and 27.78% (n = 15 strains) to erythromycin. All strains isolated from food-processing environments were resistant to oxacillin, 78.95% (n = 15) to flumenique, and 47.97% (n = 9) to sulfamethoxazole–trimethoprim.

Regarding lincomycin, most of the resistant strains were isolated from meat products (n = 73, 42.19%) and dairy (n = 8, 42.11%) (Table 2).

A total of 151 strains isolated from meat products (87.28%), 46 from seafood (85.19%), 17 from dairy products (89.47%), and 17 from food-processing environments (89.47%) were MDR (Figure 3).

## 3. Discussion

In the present study, 268 *L. monocytogenes* strains (99.63%) were resistant to at least one antibiotic. Excluding fosfomycin and flumequine, to which most isolates were inherently resistant, the overall prevalence of strains resistant to at least one antibiotic was 98.14% (N = 264). Antimicrobial resistance arises from the overuse and misuse of antimicrobials in humans and animals, as well as from insufficient infection-prevention measures [18].

As reported by Mpundu et al. [19], the worldwide combined prevalence of antibiotic-resistant *L. monocytogenes* stands at 38.1%. However, the prevalence of resistance strains differs between the world regions, ranging from 9% in Taiwan to 94.1% in South Korea [19].

In the present study, 235 strains (87.36%) showed a profile of MDR. MDR in *Listeria* was first reported in 1998 in France [1], and since then, MDR strains have been extensively isolated [20]. A high level of MDR was also observed in 2006 in North-Western Spain by Alonso-Hernando et al. (84.0%) [21] and in South Africa by Kayode et al. (82.46%) [22]. A lower prevalence (41.86%) was, however, observed in another South African study [23]. The results of the present work are of particular concern but not surprising because, within Europe, the AMR situation in Italy has repeatedly been described as the worst [18].

In the classes of antibiotics, most of the strains were resistant antibiotics belonging to the family of β-lactams (92.94%). The results are specifically alarming because β-lactams represent the treatment of choice for human listeriosis [24].

Within β-lactams, the highest level of resistance was observed against oxacillin (88.48%). Oxacillin, along with amoxicillin, are widely recognized as potent β-lactam antibiotics capable of hindering the synthesis of bacterial cell wall peptidoglycan [19]. Approximately 93% and 97% of *L. monocytogens* strains were resistant to oxacillin in the study by Harakeh et al. in Lebanon [25] and Lee et al. in Korea [26], respectively. However, a lower prevalence of resistant strains was observed in Brazil by Moreno et al. [27]. The high occurrence of oxacillin-resistant strains has implications for the clinical management and treatment strategies of listeriosis cases [15]. Of public concern are also results related to ampicillin, since approximately half of the strains were resistant to this molecule. Resistance may arise from the fact that ampicillin is one of the antibiotics used with the highest frequency because of guidelines or expert opinions [12]. Indeed, ampicillin is the first-choice treatment for listeriosis. Therefore, the high resistance observed could pose a significant risk for people, especially vulnerable groups [10]. The results are in contrast with those reported by Zhang et al. (0.4%) [28] in China and by Manyi-Loh et al. (0.0%) [23] in South Africa. However, in the study by Lee et al. [26], almost all the strains (97.0%) were resistant to this antibiotic.

A high level of resistance was also observed against fosfomycin and flumequine. This is not surprising because most strains of *L. monocytogenes* exhibit native resistance to these molecules [24,29].

Fortunately, in the present work, most of the *L. monocytogenes* strains were sensitive against gentamicin, which is currently used in combination with ampicillin or oxacillin for the treatment of human listeriosis [19].

Gentamicin, moreover, along with tetracycline, erythromycin, and trimethoprim–sulfamethoxazole, is regarded as the front-line treatment for listeriosis in animals [1]. The level of resistance observed against these antibiotics ranged from moderate (trimethoprim–sulfamethoxazole = 37.54%) to extremely low (gentamicin = 1.49%). The prevalence of resistant strains observed in the present work differs from those reported in Europe in the study by Zhang et al. [28]. The pooled prevalence reported by Zhang et al. [28] in Europe was slightly higher for tetracycline (13.0%) but lower for erythromycin (0.1%) and trimethoprim–sulfamethoxazole (8.5%). In general, the results of the present work differ from those obtained in another Italian study by Caruso et al. [30], who reported a high susceptibility of isolates to ampicillin (100%), tetracyclines (99%), erythromycin (100%), and trimethoprim–sulfamethoxazole (100%). Moreover, in the study by Caruso et al. [30], few isolates showed resistance to tetracycline (1.3%).

Moreover, no correlation has been found between resistance and serotypes (*p* > 0.05), as reported by Caruso et al. [30]. The serotypes 1/2b and 4b/4e, in addition to oxacillin, fosfomycin, and flumequine, were also highly resistant to ampicillin. Serovars 4ba and 1/2 b, along with 3b, are the major epidemic clones associated with large outbreaks as well as with sporadic human cases [31]. Instead, serovar 1/2a, frequently isolated from non-human sources [29], resulted in being highly resistant also to trimethoprim–sulfamethoxazole, probably due to the wide use of it in animal medicine [1]. Moreover, trimetoprim–sulfametoxassole is also used as second-line therapy in humans who exhibit an allergic reaction to β-lactam antibiotics [24].

No correlation in resistance/multidrug resistance (MDR) was observed between strains isolated from different sources (*p* > 0.05). Strains isolated from meat products in the present work showed a lower resistance to ampicillin compared to those observed in the study by Yucel et al. (66.0%) [32]. Regarding tetracycline, similar results were obtained in the study by Al-Nabulsi et al. [33].

Strains isolated from dairy products in the present work displayed a higher resistance to ampicillin than those reported by Rahimi, Ameri, and Momtaz (26.3%) [34], but a higher prevalence of strains resistant to tetracycline was observed in the study by Keven and Gulel (34.6%) [35].

Concerning seafood, the results of the present work differ from those reported by Jeyasanta and Patterson [36] for gentamycin (73.0% vs. 1.85%) and from Wieczorek and Osek [37] for oxacillin (57.9% vs. 90.74%).

## 4. Materials and Methods

### 4.1. Strains

In the present study, a total of 269 strains of *Listeria monocytogenes* were analyzed. Strains were isolated from meat products (n = 173), seafood (n = 54), dairy products (n = 19), sauces (n = 2), confectionary products (n = 1), ready-to-eat rice dishes (n = 1), and food-processing environments (n = 19) (Table 3). Foods were collected in the context of official controls provided by Regulation (EC) No 2073/2005 on microbiological food safety criteria and during self-monitoring controls by public or private enterprises in the Lazio region of central Italy.

Samples were transported to the laboratory within one hour and analyzed according to ISO 11290-1. In brief, 25 g portions of each sample were homogenized in 225 mL (1:10 (W/W)) of sterilized half-Fraser broth and incubated at 30 °C for 24 h. Subsequently, the incubated half-Fraser broth was (i) streaked into Oxford and Ottaviani–Agosti listeria agar incubated for 24 h at 37 °C and (ii) inoculated (0.1 mL) in 10 mL of sterilized Fraser broth incubated for 48 h at 37 °C. The enriched Fraser broth was then plated in Oxford and Ottaviani–Agosti listeria agar and incubated for 24 h at 37 °C. Presumptive colonies of listeria were biochemically identified through the API listeria^®^ bioMerieux system, and after serotyping, they were assigned to nine serovars: 1/2a, 1/2b, 1/2c, 1/a, 3a, 3b, 4b/4e, 4d/4e, and 4e (Table 3). No information regarding serovars was recorded for five isolates.

### 4.2. Antibiotic Susceptibility Testing

The antimicrobial susceptibility of the isolates was determined by the disk-diffusion method, following the EUCAST recommendations. A quality-control strain (Escherichia coli ATCC 25922) was included in the test. The following antibiotics (Oxoid, Basingstoke, UK; Becton Dickinson, Mississauga, ON, Canada) were used: ampicillin (2 µg), tetraciclin (30 µg), gentamicin (120 µg), sulfametoxassole-trimetoprim(1.25–23.75 µg), erytromycin (15 µg), meropenem (10 µg), fosfomycin (50 µg), flumequine (30 µg), lincomycin (2 µg), oxacillin (1 µg). In brief, strains were cultured twice in nutrient agar and incubated at 37 °C for 24 h. Colonies were then transferred to a brain heart infusion (BHI). The bacterial suspension in the BHI was diluted to obtain turbidity equivalent to 0.5 McFarland Standard. The contamination level was also determined by counting on Plate Count Agar (PCA; CM0325, Oxoid) after the incubation at 37 °C for 24 h. A cotton swab was used to transfer the bacterial suspension prepared in the BHI to Mueller–Hinton agar plates. Subsequently, the 10 disks containing antibiotics were added to the plates and incubated at 37 °C for 24 h. After the incubation, the diameter of the zones of inhibition was measured and compared with the breakpoint reported in Table 4.

In the evaluation of the results, the strains displaying intermediate resistance were regarded as resistant, and the strains displaying resistance to antibiotics belonging to at least three different classes were considered multidrug-resistant (MDR) [44,45].

### 4.3. Statistical Analysis

The differences in drug and multidrug resistance among the different serotypes and the different sources were assessed by a chi-squared test (χ^2^). A value of *p* < 0.05 was accepted as significant.

## 5. Conclusions

In conclusion, a high prevalence of resistant and MDR *Listeria monocytogenes* strains among the screened serotypes isolated from foods was found. The strains were highly resistant to β-lactams, the first-line drugs to treat listeriosis. The resistance against sulfamethoxazole–trimethoprim, tetracyclines, and gentamycin, largely used in cases of listeriosis, instead ranged from moderate to low. This high prevalence of *L. monocytogenes*-resistant strains in food could pose a significant risk to public health. Strict hygiene practices and antibiotic stewardship are essential to prevent the rise of antibiotic-resistant *Listeria* spp. foodborne infections. Updated studies provide insights into the effectiveness of various antibiotics against foodborne pathogens. This information is essential for healthcare professionals to prescribe appropriate treatments for infections caused by these bacteria, considering the evolving patterns of antibiotic resistance. Moreover, understanding the prevalence of antibiotic resistance in foodborne bacteria helps ensure the safety of the food supply chain. This knowledge allows for the implementation of appropriate measures to prevent contamination and the transmission of resistant bacteria through food. However, further research on the molecular profiles of *Listeria monocytogenes* strains is needed to define the association among different sources about the occurrence of resistant bacteria and resistant genes.

## Figures and Tables

**Figure 1 antibiotics-13-00525-f001:**
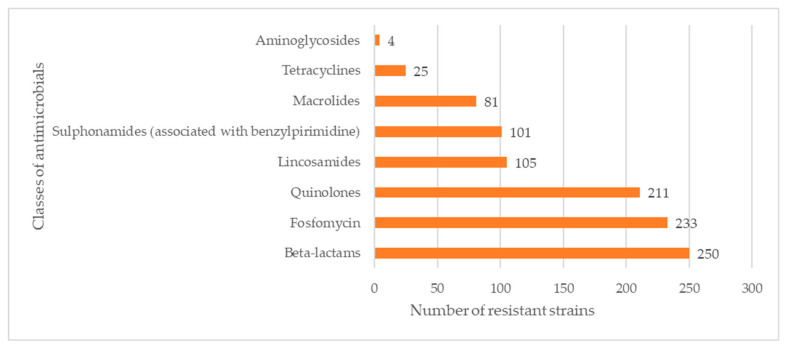
Number of *Listeria monocytogenes* strains resistant to eight classes of antimicrobials.

**Figure 2 antibiotics-13-00525-f002:**
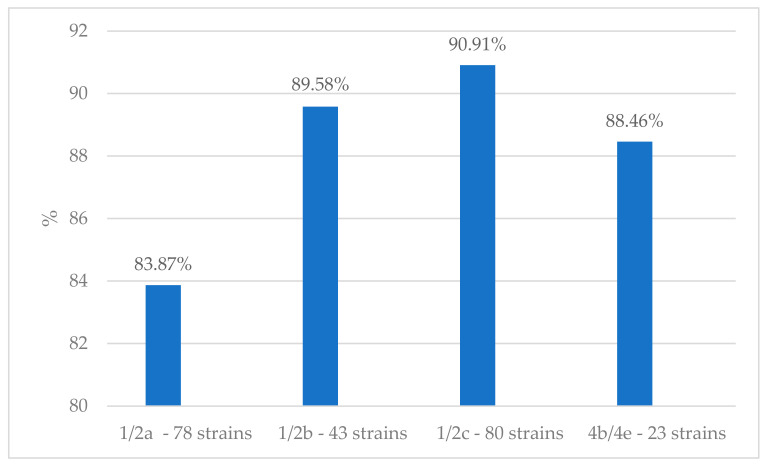
Percentage (%) of *L. monocytogens* serovars resistant to antibiotics belonging to at least three different classes (MDR).

**Figure 3 antibiotics-13-00525-f003:**
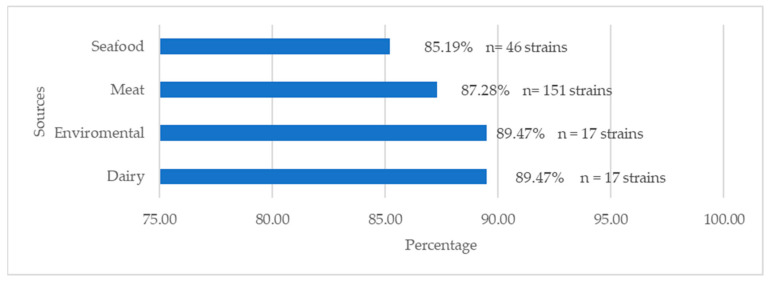
Number and percentage of *L. monocytogenes* multidrug-resistant strains isolated from dairy products, seafood, meat products, and food-processing environments.

**Table 1 antibiotics-13-00525-t001:** Number and percentage of isolates resistant to 10 antimicrobials in *Listeria monocytogenes* serovars.

	1/2a (93 Strains)	1/2b (48 Strains)	1/2c (88 Strains)	1/a (2 Strains)	3a (3 Strains)	3b (1 Strain)	4b/4e (26 Strains)	4d/4e (1 Strain)	4e (2 Strains)	Unidentified (5 Strains)	Total
Ampicillin	33.33%	54.17%	47.73%	0.00%	66.67%	0.00%	57.69%	0.00%	100.00%	60.00%	44.98%
	(n: 31)	(n: 26)	(n: 42)		(n: 2)		(n: 15)		(n: 2)	(n: 3)	(n = 121)
Lincomycin	25.81%	45.83%	48.86%	0.00%	33.33%	0.00%	38.46%	0.00%	50.00%	80.00%	39.03%
	(n: 24)	(n: 22)	(n: 43)		(n: 1)		(n: 10)		(n: 1)	(n: 4)	(n = 105)
Oxacillin	84.95%	93.75%	86.36%	100.00%	66.67%	100.00%	100.00%	100.00%	100.00%	80.00%	88.48%
	(n: 79)	(n: 45)	(n: 76)	(n: 2)	(n: 2)	(n: 1)	(n: 26)	(n: 1)	(n: 2)	(n: 4)	(n = 238)
Flumenique	80.65%	75.00%	79.55%	50.00%	66.67%	100.00%	80.77%	0.00%	100.00%	60.00%	78.44%
	(n: 75)	(n: 36)	(n: 70)	(n: 1)	(n: 2)	(n: 1)	(n: 21)		(n: 2)	(n: 3)	(n = 211)
Fosfomycin	78.49%	95.83%	88.64%	100.00%	100.00%	0.00%	80.77%	100.00%	100.00%	100.00%	85.87%
	(n: 73)	(n: 46)	(n: 78)	(n: 2)	(n: 3)		(n: 21)	(n: 1)	(n: 2)	(n: 5)	(n = 231)
Gentamycin	3.23%	0.00%	1.14%	0.00%	0.00%	0.00%	0.00%	0.00%	0.00%	0.00%	1.49%
	(n: 3)		(n: 1)								(n = 4)
Meropenem	11.83%	10.42%	15.91%	0.00%	0.00%	0.00%	11.54%	0.00%	50.00%	20.00%	13.01%
	(n: 11)	(n: 5)	(n: 14)				(n: 3)		(n: 1)	(n: 1)	(n = 35)
Erithromycin	34.41%	14.58%	37.50%	0.00%	0.00%	0.00%	11.54%	100.00%	100.00%	60.00%	30.11%
	(n: 32)	(n: 7)	(n: 33)				(n: 3)	(n: 1)	(n: 2)	(n: 3)	(n = 81)
Sulfametoxazole–thrymethoprim	45.16%	22.92%	36.36%	0.00%	33.33%	0.00%	38.46%	0.00%	100.00%	60.00%	37.54%
	(n: 42)	(n: 11)	(n: 32)		(n: 1)		(n: 10)		(n: 2)	(n: 3)	(n = 101)
Tetracycline	8.60%	8.33%	11.36%	0.00%	0.00%	0.00%	3.85%	0.00%	50.00%	20.00%	9.29%
	(n: 8)	(n: 4)	(n: 10)				(n: 1)		(n: 1)	(n: 1)	(n = 25)

**Table 2 antibiotics-13-00525-t002:** Percentage of resistance in *Listeria monocytogenes* in food products.

	Dairy Products (19 Strains)	Meat Products (173 Strains)	Seafood (54 Strains)	Confectionary Products (1 Strain)	Enviromental (19 Strains)	Sauces (2 Strains)	Ready-to-Eat Rice Dishes (1 Strain)	Total
Ampicillin	63.16 (n: 12)	43.93 (n: 76)	35.19 (n: 19)	100 (n: 1)	63.16 (n: 12)	50 (n: 1)	0.00	44.98 (n = 121)
Lincomycin	42.11 (n: 8)	42.19 (n: 73)	29.63 (n: 16)	100 (n: 1)	31.58 (n: 6)	50 (n: 1)	0.00	39.03 (n = 105)
Oxacillin	94.74 (n: 18)	85.55 (n: 148)	90.74 (n: 49)	100 (n: 1)	100 (n: 19)	100 (n: 2)	100 (n: 1)	88.47 (n = 238)
Flumenique	73.68 (n: 14)	79.19 (n: 137)	75.92 (n: 41)	100 (n: 1)	78.95 (n: 15)	100 (n: 2)	100 (n 1)	79.18 (n = 213)
Fosfomycin	84.21 (n: 16)	83.81 (n: 145)	87.04 (n: 47)	100 (n: 1)	100 (n: 19)	100 (n: 2)	100 (n: 1)	85.87 (n = 231)
Gentamicin	0.00	1.73 (n: 3)	1.85 (n: 1)	0.00	0.00	0.00	0.00	1.48 (n = 4)
Meropenem	15.79 (n: 3)	15.61 (n: 27)	5.56 (n: 3)	0.00	10.53 (n: 2)	0.00	0.00	13.01 (n = 35)
Erithomycin	21.05 (n: 4)	33.53 (n: 58)	27.78 (n: 15)	0.00	21.05 (n: 4)	0.00	0.00	30.11 (n = 81)
Sulfametoxazole–thrymethoprim	21.05 (n: 4)	39.31 (n: 68)	33.33 (n: 18)	100 (n: 1)	47.37 (n: 9)	50 (n: 1)	0.00	37.54 (n = 101)
Tetracycline	10.52 (n: 2)	10.98 (n: 19)	5.56 (n: 3)	0.00	5.26 (n: 1)	0.00	0.00	9.29 (n = 25)

**Table 3 antibiotics-13-00525-t003:** Serovars isolated for meat products (n = 173), environment (n = 54), dairy products (n = 19), seafood (n = 54), sauces (n = 2), rice dishes (n = 1), and confectionary products (n = 1).

Species	Serovar	Meat Products	Seafood	Dairy Products	Environment	Sauces	Rice Dishes	Confectionary Products	Total
*Listeria monocytogenes*	1/2a	47	27	7	9	2	1	0	93
	1/2b	19	21	6	2	0	0	0	48
	1/2c	85	0	1	2	0	0	0	88
	1/a	1	1	0	0	0	0	0	2
	3a	0	3	0	0	0	0	0	3
	3b	0	0	1	0	0	0	0	1
	4b/4e	17	1	3	4	0	0	1	26
	4d/4e	1	0	0	0	0	0	0	1
	4e	1	1	0	0	0	0	0	2
	Unidentified	2	0	1	2	0	0	0	5

**Table 4 antibiotics-13-00525-t004:** Breakpoints of the antibiotics used for the evaluation of the antibiotic susceptibility of the 269 *L. monocytogenes* strains.

Molecule	Diameter of the Halo of Inhibition (mm)			References
	Sensible	Intermediate	Resistant	
Ampicillin	≥16		<16	[38]
Meropenem	≥26		<26	[38]
Erytromycin	≥25		<25	[38]
Trimetoprim–Sulfametoxazole	≥29		<29	[38]
Tetracycline	≥19	15–18	≤14	[39]
Gentamicin	>18		<18	The breakpoints of *Staphylococcus aureus* (from EUCAST 2023 [38]) resistance were considered [40]
Oxacillin	≥18		≤17	[41]
Fosfomycin	≥24		≤24	The breakpoints of *E. coli* (from EUCAST 2023 [38]) resistance were considered [39]
Lincomycin			≤9	[42]
Flumequine	≥20	13–19	<12.5	The breakpoints of Enterobacteriales (from CLSI 2014 [43])

## Data Availability

Dataset is available on request from the authors.

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
