# Peer review of "Antimicrobial Resistance of Listeria monocytogenes Strains Isolated in Food and Food-Processing Environments in Italy"

_antibiotics, 2024, doi:10.3390/antibiotics13060525_

Round 1
Reviewer 1 Report
Comments and Suggestions for Authors The article titled "Antimicrobial Resistance of Listeria monocytogenes strains isolated in food"investigated the occurrence of Listeria monocytogenes resistant strains isolated in food.Since the antibiotic-resistance causes a public health threat and it is a very important subjectand also, the article provides some valuable data, therefore, i recommend the consideration of this piece of work after minor modification. My comments are shown in the pdf file. Thank you

no
Author Response
Dear Reviewer,
We thank you for your valuable comments.
We have made the necessary adaptations and corrections throughout the manuscript (highlighted in yellow).
A detailed answer on each remark is included in the rebuttal.
Yours sincerely,
Drs. Maria Francesca Peruzy, corresponding author
Reviewer 1:
Title
Reviewer 1: You can be more specific on the title. For example, write the Italian food or so on. In addition, the environmental samples were also carried out in your study, so, suggest to add in the title.
Authors: Thank you for this suggestion. The title has been rewritten as “Antimicrobial Resistance of Listeria monocytogenes strains isolated in food and food-processing environment in Italy “
Abstract
Reviewer 1: investigated
Authors: Thank you for this suggestion. The word has been rewritten; please refer to line 20.
Reviewer 1: Please specify what kind of food and how many samples?
Authors: Thank you for this suggestion. The kinds of foods and the number of samples has been added to the abstract. Please refer to lines 21- 22.
Reviewer 1: Italic
Authors: Sorry for the mistake. The word is now in italics, please refer to line 25.
Keywords
Reviewer 1: Suggest to add "public health". You can have up to 6 keywords.
Authors: Thank you for this suggestion. The keyword “public health” has been added.
Introduction
Reviewer 1: Your study is on the antimicrobial resistance and consecutive public health concerns. So, expecting more elaboration in the introduction.
Authors: Thank you for pointing this out. The introduction has been redesigned according to reviewers' suggestions.
Reviewer 1: There should be a statement of the problem before the objective. Please mention what is the gap and how your study is filling it
Authors: Thank you for this suggestion. The introduction has been redesigned according to reviewers' suggestions.
Results
Reviewer 1: Fig. 1 Please insert the title for X axis
Authors: Thank you for this suggestion. The title for axis X has been added in Figure 1.
Reviewer 1: Table 1 Please define the units properly
Authors: thank you for the suggestion, the units have been added to Table 1.
Reviewer 1: The numbers are 11 but the title is 10
Authors: The antimicrobial are 10. The authors agree with the reviewer that the table 1 could generate misunderstanding and therefore it has been modified.
Reviewer 1: Figure 2. Title should be below the Figure. Please also add the axis title.
Authors: the caption has been moved below the Figure 2.
Reviewer 1: Table 2 food products
Authors: the word “food” has been added in Table 2.
Reviewer 1: Red meat you mean?
Authors: meat product samples included all kinds of meat.
Reviewer 1: Figure 3. Title below the Figure
Authors: the caption of Figure 3 has been moved below.
Discussion
Reviewer 1: Italic
Authors: following the reviewer's suggestion, the first paragraph has been removed.
Conclusion
Reviewer 1: Italic
Authors: Sorry for the mistake, italic has been now used. please refer to the line 328.
Reviewer 1: Please write about the findings first and implications of your study. In the end, write the conclusion.
Authors: thanks to the reviewer’s suggestion the conclusion has been modified, please refer to the lines 330-335.
Reviewer 1: Please also write about the future studies and trends.
Authors: thanks to the reviewer’s suggestion the conclusion has been modified, please refer to the lines 342-344.
Reviewer 2 Report
Comments and Suggestions for Authors
Antibiotics resistant microbes are a major concern for worldwide public health, partly due to their extensive usage in both people and animals. Since the research were rise in title of “Antimicrobial Resistance of Listeria monocytogenes strains isolated in food”. The research work is well-written and arranged by the authors. I agree with the research article and don't generally have any major concerns. Several of the things that have been noticed, in my opinion, are not specified as follows.
- One major idea or heading theme should typically be stated in a single paragraph. The authors may be thoroughly cited in the introduction, such the order of L37–L39 might be rearranged. The author may have revised the information contained in this section with novel findings. L77 in text citation could be revised.
- Rechecking the terminology of Amynoglicosides in figure 1.
- Materials and methods, it was noted that the author needed to provide additional details and do more of this clearly.
o Why did not the number of specimens obtained correspond similarly to serovar for each of the six categories of food samples?
- Please follow the journal guide's instructions for double-checking the references.
Author Response
Dear Reviewer,
We thank you for your valuable comments.
We have made the necessary adaptations and corrections throughout the manuscript (highlighted in yellow).
A detailed answer on each remark is included in the rebuttal.
Yours sincerely,
Drs. Maria Francesca Peruzy, corresponding author
Reviewer 2
Reviewer 2: One major idea or heading theme should typically be stated in a single paragraph. The authors may be thoroughly cited in the introduction, such the order of L37–L39 might be rearranged. The author may have revised the information contained in this section with novel findings. L77 in text citation could be revised.
Authors: thanks to the reviewer’s suggestion the authors have rearranged line 37-40. Moreover, the introduction has been redesigned according to reviewers' suggestions. The citation at line 81 has been corrected.
Reviewer 2: Rechecking the terminology of Amynoglicosides in figure 1.
Authors: thanks to the reviewer’s suggestion the authors have re-written the name of the class of the antibiotic correctly
Reviewer 2: Materials and methods, it was noted that the author needed to provide additional details and do more of this clearly. Why did not the number of specimens obtained correspond similarly to serovar for each of the six categories of food samples?
Authors: Additional details have been added to Material and Methods. Moreover, concerning seafood a little mistake was made in the caption of Table 3. Sorry for this. The mistake has now been corrected. Please refer to lines 277-282 and 307-316.
Reviewer 3 Report
Comments and Suggestions for Authors
I congratulate the authors for the chosen topic, which is of major importance for both veterinary and human medicine. There are hard-earned results that once again reveal the danger of MDR bacterial strains.
I have only a suggestion-the introduction could easily be restructurated.
Good luck in the future.
Author Response
Dear Reviewer,
We thank you for your valuable comments.
We have made the necessary adaptations and corrections throughout the manuscript (highlighted in yellow).
A detailed answer on each remark is included in the rebuttal.
Yours sincerely,
Drs. Maria Francesca Peruzy, corresponding author
Reviewer 3
I congratulate the authors for the chosen topic, which is of major importance for both veterinary and human medicine. There are hard-earned results that once again reveal the danger of MDR bacterial strains. I have only a suggestion-the introduction could easily be restructurated. Good luck in the future.
Authors: Thank you. The introduction has been restructured according to reviewers' suggestions.
Reviewer 4 Report
Comments and Suggestions for Authors
I have read with interest the manuscript entitled “ Antimicrobial Resistance of Listeria monocytogenes strains isolated in food” by Antonio et al, which contributes to our understanding of antimicrobial resistance in L. monocytogenes. However, certain revisions are required before the manuscript can be considered for publication.
Title and Abstract
Well written, no comment
Introduction
The introduction seems well written; however, the epidemiological context, such as transmission and antimicrobial resistance patterns on L. monocytogens, isn't clearly stated. In addition, what has been done in Italy, Europe, or elsewhere in the world on this topic and what is the critical gap that the current work is trying to fill aren't clear. I think there are several works on antimicrobial resistance of L. monocytogens. An example of these are given below for your further reference:
1. https://www.mdpi.com/2076-2607/12/5/954
2. https://www.mdpi.com/2079-6382/11/10/1447
3. https://bmcmicrobiol.biomedcentral.com/articles/10.1186/s12866-021-02335-7
Please revise the introduction to indicate the novelty and the specific research gap that this study is trying to address.
Materials and Methods
- On what basis was the list of antimicrobials tested chosen to be included in the study? The antibiotic susceptibility testing section lacks a rationale for choosing the specific antibiotics tested.
- The sampling condition, sampling method and specific geographic location where the samples were collected isn't clear.
Results
- The presentation of data in figures, tables and numeric values in brackets needs to be more consistent. For instance, Figure 1 should have a more descriptive title and legend to ensure clarity.
- The text in the results section does not describe the statistical significance of differences in resistance between serotypes and sources.
- In multiple paragraphs, results are expressed using the word most ( for example, line 144). Consider updating the text to make it more specific.
Line 90: sensible or sensitive? Does the number in brackets show a number of isolates or percentages?
Lines 137-140: Revise the paragraph to make it consistent
Discussion
- There is a need for a more critical analysis of the potential reasons for observed resistance patterns, such as the included antibiotics routinely used to treat infections in humans in the study area, the role of agricultural practices, the use of antibiotics in veterinary medicine, and environmental factors.
- The discussion should better add the implications of the findings for public health and food safety.
· Line 157: Does it refer to antimicrobial treatment or antimicrobial chemotherapy?
· The first paragraph of the discussion doesn’t seem to benefit the discussion and can be removed.
Author Response
Dear Reviewer,
We thank you for your valuable comments.
We have made the necessary adaptations and corrections throughout the manuscript (highlighted in yellow).
A detailed answer on each remark is included in the rebuttal.
Yours sincerely,
Drs. Maria Francesca Peruzy, corresponding author
Reviewer 4
Title and Abstract
Reviewer 4: Well written, no comment
Authors: Thank you .
Introduction
Reviewer 4: The introduction seems well written; however, the epidemiological context, such as transmission and antimicrobial resistance patterns on L. monocytogens, isn't clearly stated. In addition, what has been done in Italy, Europe, or elsewhere in the world on this topic and what is the critical gap that the current work is trying to fill aren't clear. I think there are several works on antimicrobial resistance of L. monocytogens. An example of these are given below for your further reference:
- https://www.mdpi.com/2076-2607/12/5/954
- https://www.mdpi.com/2079-6382/11/10/1447
- https://bmcmicrobiol.biomedcentral.com/articles/10.1186/s12866-021-02335-7
Please revise the introduction to indicate the novelty and the specific research gap that this study is trying to address.
Authors: Thank you for your suggestion, the introduction has been modified and the suggested citations have been included.
Materials and Methods
Reviewer 4: On what basis was the list of antimicrobials tested chosen to be included in the study? The antibiotic susceptibility testing section lacks a rationale for choosing the specific antibiotics tested.
Authors: The list of the antibiotics used in the present work was chosen considering the EUCAST guidelines (eucast: Clinical breakpoints and dosing of antibiotics) and the literature reported in table 4.
Reviewer 4: The sampling condition, sampling method and specific geographic location where the samples were collected isn't clear.
Authors: Detailed information have been provided in material and methods. Please refer to lines 277-282.
Reviewer 4: The presentation of data in figures, tables and numeric values in brackets needs to be more consistent. For instance, Figure 1 should have a more descriptive title and legend to ensure clarity.
Authors: The caption of each figure/table has been modified
Reviewer 4: The text in the results section does not describe the statistical significance of differences in resistance between serotypes and sources.
Authors: Thank you for pointing this out. The statistical results have now been included also in the results section. Please refer to lines 140-141.
Reviewer 4: In multiple paragraphs, results are expressed using the word most ( for example, line 144). Consider updating the text to make it more specific.
Authors: The exact percentage has now been included in the results. Please refer to lines 133-134. Moreover, the paragraph in lines 189-191 has been modified as “A total of 151 strains isolated from meat products (87.28 %), 46 from seafood (85.19 %), 17 from dairy products (89.47 %), and 17 from food-processing environment (89.47 %) were MDR (Figure 3). “
Reviewer 4: Line 90: sensible or sensitive? Does the number in brackets show a number of isolates or percentages?
Authors: Thank you for pointing this out. These were indeed mistakes. The mistakes have now been corrected. Please refer to lines 107.
Reviewer 4: Lines 137-140: Revise the paragraph to make it consistent
Authors: The paragraph has been improved. Please refer to lines 161-170.
Discussion
Reviewer 4: There is a need for a more critical analysis of the potential reasons for observed resistance patterns, such as the included antibiotics routinely used to treat infections in humans in the study area, the role of agricultural practices, the use of antibiotics in veterinary medicine, and environmental factors.
Authors:
Thank you for pointing this out. Potential reasons for the observed resistance pattern have been included in the introduction.
Reviewer 4: The discussion should better add the implications of the findings for public health and food safety.
Authors: The implications of the findings and food safety have now been added to the conclusion. Please refer to lines 330-334.
Reviewer 4: Line 157: Does it refer to antimicrobial treatment or antimicrobial chemotherapy?
Authors: The first paragraph has been removed as suggested by the reviewer.
Reviewer 4: The first paragraph of the discussion doesn’t seem to benefit the discussion and can be removed.
Authors: The first paragraph has been removed as suggested by the reviewer.